# Pro- and Anti-Inflammatory Cytokines in the First Trimester—Comparison of Missed Miscarriage and Normal Pregnancy

**DOI:** 10.3390/ijerph18168538

**Published:** 2021-08-12

**Authors:** Maciej Kwiatek, Tomasz Gęca, Anna Kwaśniewska

**Affiliations:** Chair and Department of Obstetrics and Pathology of Pregnancy, Medical University of Lublin, 20-059 Lublin, Poland; tomgeca@wp.pl (T.G.); anna.kwasniewska@umlub.pl (A.K.)

**Keywords:** missed miscarriage, proinflammatory cytokines, anti-inflammatory cytokines, immune response

## Abstract

The advantage in response of Th2 over Th1 is observed in normal pregnancy in peripheral blood. A disturbance of this balance can lead to symptoms of miscarriage and pregnancy loss. The aim of this study was to evaluate the pro- and anti-inflammatory cytokines in sera of women who were diagnosed with missed miscarriage in the first trimester and to compare this systemic immune response to the response in women with normal pregnancy. The study group consisted of 61 patients diagnosed with missed miscarriage. In total, 19 healthy women with uncomplicated first trimester created the control group. Cytokines were determined in the maternal serum by ELISA. The analysis included INF-γ, TNF-α, Il-1β, Il-4, Il-5, Il-6, Il-9, Il-10, Il-13 and TGF-β1. Th1 cytokine levels in the study group reached slightly higher values for INF-γ, Il-1β and slightly lower for IL-6 and TNF-α. In turn, Th2 cytokine levels in the study group were slightly higher (Il-9, Il-13), significantly higher (Il4, *p* = 0.015; Il-5, *p* = 0.0003) or showed no differences with the control group (Il-10). Slightly lower concentration involved only TGF-β1. Analysis of the correlation between levels of pro- and anti-inflammatory cytokines resulted in some discrepancies, without showing predominance of a specific immune response. The results did not confirm that women with missed miscarriage had an advantage in any type of immune response in comparison to women with normal pregnancy.

## 1. Introduction

During pregnancy, the function of the maternal immune system is to protect the mother and the fetus from pathogens, as well as prevent the rejection of the allogeneic transplant of the fetal–placental unit [1]. Paternal MHC class I antigens (HLA-C) on the surface of trophoblast cells are presented by antigen presenting cells (APCs) to maternal helper T cells [2]. As a result of activation, lymphocytes are able to secrete various cytokines with pro- and anti-inflammatory properties [3]. Th1 cells secrete mainly INF-γ, but also Il-1β, Il-2, Il-12, Il-15 and TNF-α [3]. Th2 cells produce mainly Il-10, as well as Il-4, Il-5, Il-6, Il-9, Il-13, TGF-β1 [3]. Th1 cytokines participate in the late hypersensitivity reaction and direct the cellular response, while Th2 cytokines activate B cells, thereby enhancing the humoral response [4].

Among Th2 cytokines, Il-10 is immunosuppressive to Th1 cells. Il-10 acts primarily on APCs that promote T-cell anergy and stimulate regulatory T cells secreting Il-10 and TGF-β [5,6]. Excessive activation of Th1 cells negatively affects the number and function of regulatory T cells and increases the population of Th17 and NK cells in both peripheral blood and decidua. Such data were obtained in women with recurrent miscarriages [7,8,9,10,11]. In contrast, peripheral blood mononuclear cells have been shown to secrete more Th1 than Th2 cytokines in women with normal first trimester of pregnancy compared to women with spontaneous miscarriage or a history of recurrent miscarriages that occurred subsequently [12,13]. The Th1 cytokines tested included Il-2, TNF α and β, INF-γ, and the Th2 cytokines included Il-4, Il-5, Il-6 and Il-10. Although similar methods (ELISA and flow cytometry) were used in the studies, the results of the publication remain contradictory to the theory of a generalized inflammatory response of the body in women with early pregnancy loss. It should be noticed that cytokines are produced not only by Th cell subtypes. APCs cells and other immune cell populations are also involved in their synthesis and secretion [14]. Therefore, the assessment of the concentration of cytokines in peripheral blood serum seems to best reflect the body’s immune status. In addition, the complex relationships, synergistic or antagonistic mechanism of action of individual cytokines in relation to each other may cause difficulty in interpreting the results of the studies.

The aim of this study was to evaluate the pro- and anti-inflammatory cytokines in sera of pregnant women who were diagnosed with missed miscarriage in the first trimester and to compare this systemic immune response to the response of women with normal pregnancy.

## 2. Material and Methods

The study was conducted among patients of the Department of Obstetrics and Pregnancy Pathology at the Medical University in Lublin, hospitalized due to missed miscarriage before 13 weeks of pregnancy, and among healthy pregnant women with normal course of the first trimester of pregnancy. Gestational age was determined by the last menstrual period and corrected by ultrasound when the last menstrual period was unknown or in the case of irregular periods. Missed miscarriage was diagnosed based on the ultrasound examination, in which fetal heart rate was not detected after 6 weeks of pregnancy or an empty gestational sac was displayed. At CRL (crown-rump length) below 20 mm, the diagnosis was confirmed by at least two determinations of chorionic gonadotropin concentration. Patients with uterus abnormalities, autoimmune diseases, infection symptoms, and abnormal karyotype of a woman or her partner were excluded from the study. The research was approved by the Bioethical Committee operating at the Medical University of Lublin. Each participant received and signed an informed consent form for voluntary participation in the study.

A total of 80 women were qualified for the study. The study group consisted of 61 patients diagnosed with missed miscarriage and 19 healthy women with uncomplicated first trimester who created the control group. The study material was venous blood collected into S-Monovette tubes with EDTA as anticoagulant (Sarstedt, Nümbrecht, Germany). Blood in the study group was collected prior to the administration of prostaglandins that were applied in order to induce a miscarriage in cases of stillbirth or blighted ovum or before the curritage of the uterus if prostaglandins were not used. Blood in the control group was collected during routine tests during the first trimester. Within 30 min of blood collection, the samples were centrifuged (Sigma 1-6P, Polygen) for 10 min at a speed of 2800 rpm. The obtained plasma was aliquoted in 200 μL in Eppendorf tubes (Medlab Products, Darmstadt, Germany) and stored at −75 °C (Platinum Angelantoni 500, Perugia, Italy) until assayed. Serum proinflammatory and anti-inflammatory cytokines were determined using commercial ELISA kits (Enzyme-linked Immunosorbent Assay) based on immune responses. Determination of the concentration of tested cytokines was carried out strictly according to the recommendations given by the kit manufacturers. Levels of Il-4, Il-6, Il-10, IFNγ and TNFα were determined using reagent kits from Diaclone SAS, a company of the group Biotech Investissment (Besancon, France). Il-5 and Il-9 concentrations were determined using reagent kits from FineTest, Wuhan Fine Biological Technology Co., Ltd. (Wuhan, China). Serum Il-1β, Il-13 and TGF-β1 were determined by the test of DRG Instruments GmbH (Marburg, Germany).

## 3. Statistical Analysis

Statistica 13 PL and MedCalc 15.8 were used for statistical analysis. Results *p* < 0.05 were considered statistically significant. The Chi2 (χ^2^) independence test was used to compare the distribution of demographic and clinical variables in the control and study groups. Due to the non-normal distribution of all studied cytokines (assessed with use of D’Agostino-Pearson test), nonparametric tests were used. The non-parametric U-Mann-Whitney test (with two variables, continuous data) was used to analyze the differences in the values of the examined demographic and clinical variables of continuous nature and cytokines. Analysis of ANOVA covariance was used to assess the possible influence of covariates on obtained results. The non-parametric Spearman rank correlation test was used to analyze the correlation between selected continuous variables or variables whose distribution can be represented on the ordinal scale.

## 4. Results

### 4.1. Characteristics and Comparison of Selected Demographic and Clinical Variables in the Study and Control Groups

The study group included 61 women diagnosed with miscarriage in the first trimester of their current pregnancy. The mean age was about 30 years (SD ± 4.99). The median body weight was 67 kg. The median BMI was 23.59 (range: 17.63–41.27). Most women (over 60%) gave birth once and had no history of miscarriage (86.89%). The control group consisted of 19 women. The mean age was about 27 years (SD ± 3.05), the median body weight was 58 kg. The median BMI was 21.23 (range: 16.70–31.96). In the study group, women were significantly older (30.35 vs. 27.26 years; *p* = 0.0086), had higher body weight (67.00 vs. 58.00 kg; *p* = 0.0291) and higher BMI (23.59 vs. 21.23 kg/cm^2^; *p* = 0.0385). However, analysis of covariance showed, that the only significant difference between the study and control group was age (*p* = 0.0466). Detailed data are included in Table 1.

### 4.2. Comparison of Values of Selected Cytokines in the Study and Control Groups and in the Subgroups of the Test Group

The concentrations of tested cytokines in the study and control groups were compared. A statistically significant difference was found between the Il-5 (30.19 vs. 16.65, respectively, *p* = 0.0003) and Il-4 (0.64 vs. 0.24, respectively, *p* = 0.0155) between the study and control group. However, when analysis of covariance was performed, that the only significant difference between the study and control group was noted in the case of IL-5 (*p* = 0.0008). Detailed data are included in Table 2 and Figure 1.

Correlations between selected demographic and clinical variables and cytokines assessed in blood serum in the study group were examined. There was a weak negative correlation between INF-ɣ concentration and the number of miscarriages (rho = −0.296, *p* = 0.0204, Figure 2). Weak positive correlations were found between Il-5 and the number of miscarriages (rho = 0.264, *p* = 0.0394; Figure 3). Detailed data on Spearman’s rank correlation between selected demographic and clinical variables and cytokines in the study group are provided in Table 3.

### 4.3. Correlations between Pro- and Anti-Inflammatory Cytokines

Correlations between pro-and anti-inflammatory cytokines evaluated in blood serum in the control group were investigated. A positive strong correlation was found between Il-4 and Il-1ß (rho = 0.731, *p* = 0.0004). In addition, there was a moderate negative correlation between Il-10 and Il-6 (rho = −0.610, *p* = 0.0055) as well as a moderate positive correlation between Il-10 and TNF-α (rho = 0.500, *p* = 0.0294). Detailed data on Spearman’s rank correlation between pro- and anti-inflammatory cytokines evaluated in blood serum in control women are provided in Table 4.

Correlations between pro- and anti-inflammatory cytokines assessed in blood serum in the study group were examined. A positive correlation was found between Il-4 and INF-γ (weak correlation, rho = 0.395, *p* = 0.0016), Il-1ß (moderate correlation, rho = 0.616, *p* < 0.0001), TNF-α (strong correlation, rho = 0.826, *p* < 0.0001). In turn, Il-5 positively correlated with TNF-α (moderate correlation, rho = 0.448, *p* = 0.0003). Il-9 correlated positively with Il-1ß (rho = 0.298, *p* = 0.0198) and TNF-α (moderate correlation, rho = 0.517, *p* < 0.0001) and negatively with Il-6 (weak correlation, rho = −0.286, *p* = 0.0253). In addition, Il-10 positively correlated with INF-γ (weak correlation, rho = 0.276, *p* = 0.0313), Il-6 (weak correlation, rho = 0.259, *p* = 0.0436), TNF-α (moderate correlation, rho = 0.499, *p* < 0.0001). Il-13 positively correlated with INF-γ (weak correlation, rho = 0.375, *p* = 0.0029), Il-1ß (moderate correlation, rho = 0.469, *p* = 0.0001), TNF-α (weak correlation, rho = 0.366, *p* = 0.0037). TGF-ß1 correlated positively with TNF-α (weak correlation, rho = 0.359, *p* = 0.0045). Detailed data on Spearman’s rank correlation between pro- and anti-inflammatory cytokines assessed in blood serum in the study group are provided in Table 5.

## 5. Discussion

Achieving maternal immune tolerance is critical to the development of early pregnancy [15]. The local proinflammatory environment is needed during implantation and for the needs of angiogenesis in the uterus [16]. However, in normal pregnancy in peripheral blood, the advantage of Th2 response over Th1 is observed, which is a significant phenomenon of pregnancy, and disturbance of this balance can lead to symptoms of miscarriage and pregnancy loss [17]. Miscarriages affect about 15% of clinically confirmed pregnancies, but the real percentage is probably much higher [18]. It has been proven so far that during pregnancy, the immune response increases, which can additionally increase in miscarriage [19]. The methodology based on assessing the concentration of cytokines in peripheral blood and not only in isolated peripheral blood mononuclear cells is reflected in other publications [19,20,21,22] and seems to best reflect the body’s immune status. In missed miscarriages, both an increased proinflammatory response understood as an increase in serum Th1 cytokine levels and lower levels of Th2 cytokines have been described [15,23,24].

Analysis of the results of our work does not clearly indicate the predominance of Th1 or Th2 cytokines in the study or control group. The serum concentrations of cytokines did not depend on the number of pregnancies, parturition or gestational age at which missed miscarriage was diagnosed. Th1 cytokine levels in women with miscarriage compared to the control group reached slightly higher values for INF-γ, Il-1β and slightly lower for Il-6 and TNF-α. In turn, Th2 cytokine levels in the study group were slightly higher (Il-9, Il-13), significantly higher (Il-4, Il-5) or showed no differences with the control group (Il-10). Slightly lower concentration concerned only TGF-β1. Analysis of the correlation between the levels of pro- and anti-inflammatory cytokines also resulted in some discrepancies without showing a predominance of a specific immune response.

Obesity is associated with chronic systemic inflammation. It is due to local immune responses in visceral adipose tissue. The studies report that obesity is associated with significantly elevated levels of Il-5, Il-10, Il-12, Il-13, IFN-γ and TNF-α [25]. In our study group we observed significantly higher maternal body mass and BMI. We found positive correlation between all investigated cytokines except Il-10, which may confirm up-regulation of certain pro- and anti-inflammatory cytokines in obesity. Maternal age is another variable which may influence cytokine profile. Older maternal age is related to higher risk of miscarriage. It was proved that aging is associated with increased inflammatory activity reflected by increased circulating levels of TNF-α, Il-6, cytokine antagonists and acute phase proteins [26]. However, in our study, we did not find such correlation and our data are even contradictory as TNF-α and Il-1ß had significant negative correlation with maternal age.

In the Calleja-Agius et al. study comparing pregnant and non-pregnant women, it was found that in the first group higher levels of tested proinflammatory cytokines were found, both in serum and intracellularly [19]. Th1 cytokine levels were higher in pregnancies ending in spontaneous abortion than in normal pregnancies. Notably, the authors examined abortive concept products for karyotype and did not show any significant differences in the levels of Th1 or Th2 cytokines tested in euploid and aneuploid pregnancies.

Lower plasma INF-γ concentrations may indicate a role of that substance in active labor [27]. To date, many authors have reported significantly higher levels of INF-γ in spontaneous or recurrent miscarriages, which appears to be the most important Th1 type cytokine [15,19,28,29]. Used in vitro, it inhibits the development of trophoblast, and its administration to pregnant mice results in miscarriage [30]. In contrast, Bates et al. showed no significant difference in INF-γ concentration in women with recurrent miscarriage, and our analysis showed a negative correlation with the number of miscarriages experienced [13].

Production of Il-1β in pregnancy is stimulated by β-hCG [31]. The cytokine affects the implantation process by affecting the endometrium at the implantation site [32]. It enhances the secretion of PGE2 and LIF by maternal tissues and stimulates the expression of the β3 integrin subunit. It is probably responsible for regulating placental growth [33]. There are discrepancies in the literature regarding the role of Il-1β in the pathogenesis of miscarriages. Pei-Yan et al. describe a significant increase in its concentration in recurrent miscarriages [34]. On the other hand, in animal model studies it was found that low concentrations of Il-1β receptor antagonist before embryo transfer resulted in a lower percentage of successful pregnancies [35].

Il-6 acts multidirectionally, because on one hand it stimulates inflammatory processes and on the other hand, it also suppresses TNF-α secretion or stimulates B cell differentiation into plasma cells [36]. This versatility results in different qualifications of this cytokine by the researchers. Some present it as anti-inflammatory and submit the results referring to the relationship with Th1 cytokines [19,24]. Others present the results showing its strong proinflammatory properties by contrasting it with Th2 cytokines [34]. In our work, it was true that no statistical differences in serum Il-6 concentration were found in the study and control group, but in correlation with Il-10 it was found that in missed miscarriage, the increase in Il-10 concentration was followed by a significant increase in Il-6 concentration. In healthy pregnant women the opposite phenomenon was observed. Some publications confirm significantly higher plasma Il-6 levels in missed miscarriage [19] and in recurrent miscarriages [34] compared to normal pregnancies. Analysis of the relationship between cytokines and human chorionic gonadotropin showed significantly higher Il-6/β-hCG ratios in women with spontaneous abortion [37]. In turn, Paradisi et al. showed a completely different relationship, finding significantly higher concentrations of Il-6 in normal pregnancies and in threatening miscarriages compared to missed miscarriage [24].

In most publications on spontaneous and recurrent miscarriages, serum TNF-α levels were significantly higher in the studied groups than in women with normal pregnancy [15,23,37]. Freis et al. also described a significantly higher TNF-α/hCG ratio in spontaneous abortion [37]. Pei-Yan et al. found slightly higher TNF-α levels in women with recurrent early pregnancy losses, but showed a significantly higher TNF-α/TGF-β1 ratio in this group of women [34]. In turn, in the work of Yamada et al. lower TNF-α levels were found in women with recurrent miscarriages [38]. In our studies, all Th2-type cytokines correlated positively with TNF-α. Interestingly, in the control group, the correlations between Il-10 and TNF-α showed the same positive relationship as in the study group.

The conclusions drawn from studies on missed miscarriages are often based on calculated ratios expressing the ratio of individual cytokine concentrations [15,23,24]. Particular attention is paid to Il-10, which is responsible for maintaining the Th2-type immune response when it is stabilized in the maternal body [39]. Interestingly, the systemic down-regulation of several members of the Il-10 family of cytokines plays an important role in the activation of myometrial smooth cells associated with uterine contractions during active labor [27]. The concentration ratio of TNF-α/Il-10 and INF-γ/Il-1 achieves lower values of these coefficients in the case of spontaneous miscarriages in the first trimester [19]. This is in contrast to the overall Th1/Th2 cytokine relationship in miscarriages, but the authors discuss the oxidative stress that occurs in the event of miscarriage and subsequent compensatory anti-inflammatory response in maternal blood. Normal pregnancies have higher Il-10 levels compared to women with recurrent miscarriages [15,29]. However, among women with a history of recurrent miscarriages, whether the next pregnancy is successful or the next miscarriage occurs, Il-10 levels are comparable [29]. In turn, Bates et al. did not show differences in Il-10 concentration between the group of recurrent miscarriages and normal pregnancies [13]. In our studies, both Il-10 concentrations and correlations between Il-10 and TNF-α and INF-γ were comparable in the study and control groups.

We observed significantly higher levels of Il-4 and Il-5 (which belong to Th2 cytokines) in women with miscarriage. High Il-4 concentrations are reflected in the work of Pei-Yan et al. based on the material of women with recurrent pregnancy losses [34]. The authors claim that the increase in Il-4 concentration does not change the proinflammatory polarization of cytokines in the group of women with miscarriage, as it is accompanied by a significant increase in INF-γ concentration and the INF-γ/Il-4 ratio remains twice as high as in the control group. We showed a positive correlation of Il-4 with both INF-γ as well as with Il-1β and TNF-α in the case of miscarriage. These findings remain consistent with previous reports [34,40]. On the other hand, Marzi et al. [41] observed that in normal pregnancy, lower levels of INF-γ are accompanied by a significant increase in Il-4 and Il-10, which, however, was not confirmed in our work. Other reports show no difference in Il-4 levels in women with spontaneous miscarriage, recurrent miscarriage and normal pregnancy [13,29]. Il-5 has not yet been studied in missed miscarriages. We found a positive correlation of this cytokine with the number of miscarriages. However, according to the literature, Il-5 reaches significantly lower serum levels in women with recurrent miscarriage than in women at the end of the normal first trimester and before delivery [15,23]. Makhseed et al. assessed the concentration of cytokines by the ELISA method [23]. They showed that the concentrations of Il-5 as well as Il-4 or TNF-β are undetectable in sera of both healthy pregnant women and those with recurrent miscarriage.

Concentrations of other Th2 cytokines determined as follows: Il-9, Il-13 and TGF-β1 did not show statistical differences in both groups. Similar conclusions regarding Il-13 were presented by Hossein et al. [29]. In turn, Paradisi et al. did not show differences in Il-13 concentration in women with threatening miscarriage, normal and non-pregnant pregnancy. However, in missed miscarriage these values were significantly lower than in these three groups [24]. TGF-β is a key factor controlling trophoblast invasion and growth. It is probably an essential element in the regulation of cytokine network, and is responsible for the state of immune tolerance in pregnancy and can maintain the allogeneic fetus [29,37]. Among other aspects, its low concentrations in peripheral blood mononuclear cells in miscarriages and biochemical pregnancy have been described [42]. The results of reports on recurrent miscarriages, however, remain contradictory, because they present either no difference [29], significantly higher levels [43] or significantly lower levels [34] of TGF-β in the plasma of these women compared to healthy pregnant women. Pei-Yan et al. additionally found a negative correlation with Il-6, INF-γ, Il-1β and TNF-α [34], but these relationships were not found in our work. Il-9 reaches the highest concentrations of all cytokines and chemokines both in pregnant and non-pregnant uterus and additionally in the placenta, but there are no reports of values that may be achieved in the serum of women with miscarriage [44].

While we have not clearly shown which type of immune response dominates in missed miscarriage in the first trimester of pregnancy, the vast number of different relationships in the cytokine network should be noted. Cytokines circulating in the maternal bloodstream perfectly reflect her immune status, but other significant changes also occur in the tissues of the pregnant uterus itself, which was not included in our research. Since the maternal blood is much more accessible than endometrial lining (the place of implantation of the embryo), in the future the immunological diagnosis of spontaneous abortions can be based on such material. It should also be remembered that miscarriages are a heterogeneous group of cases with very different etiologies. However, as presented in the cited publications, the most common cause, i.e., genetic background, does not affect the results of immunological tests in a significant way [19,45]. The commonness of spontaneous miscarriages, and their important psychological and economic aspects, makes this issue the point of attention and deepening knowledge about processes that could lead to the loss of early pregnancy is necessary.

## 6. Conclusions

In conclusion, the results presented here did not confirm that women with missed miscarriage compared to women with normal pregnancy had an advantage of any type of immune response. The isolated higher levels of Il-4 and Il-5 and the positive correlations between most of the Th1 and Th2 cytokines tested may be an expression of an overall enhanced immune response after fetal death in the first trimester of pregnancy. This issue requires further research.

## Figures and Tables

**Figure 1 ijerph-18-08538-f001:**
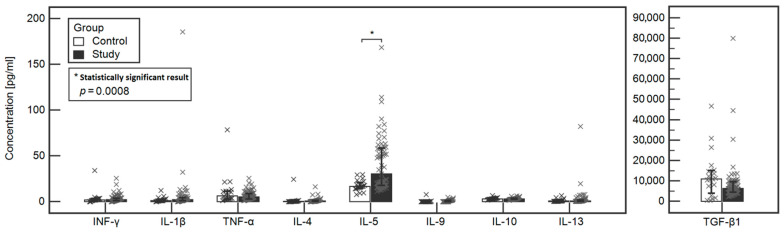
Comparison of the levels of tested cytokines between the study and control groups.

**Figure 2 ijerph-18-08538-f002:**
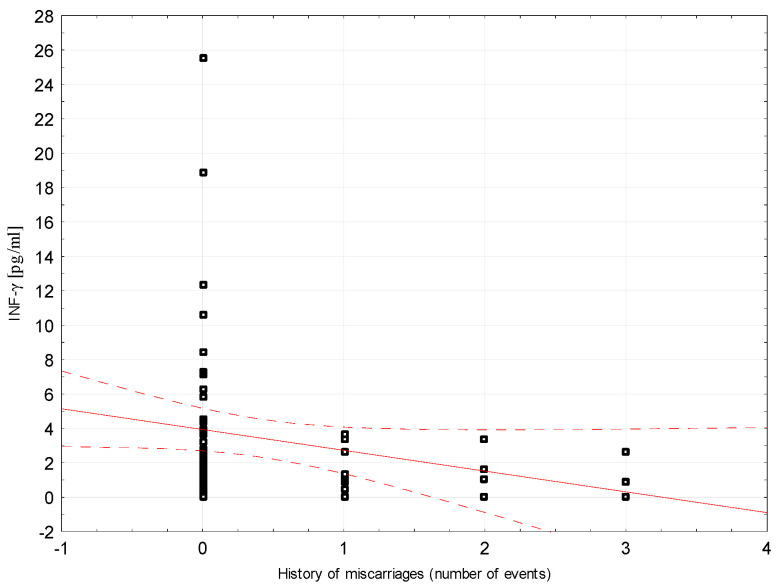
A scatter graph showing the correlation between the number of miscarriages and the INF-γ concentration assessed in the blood serum of patients from the study group.

**Figure 3 ijerph-18-08538-f003:**
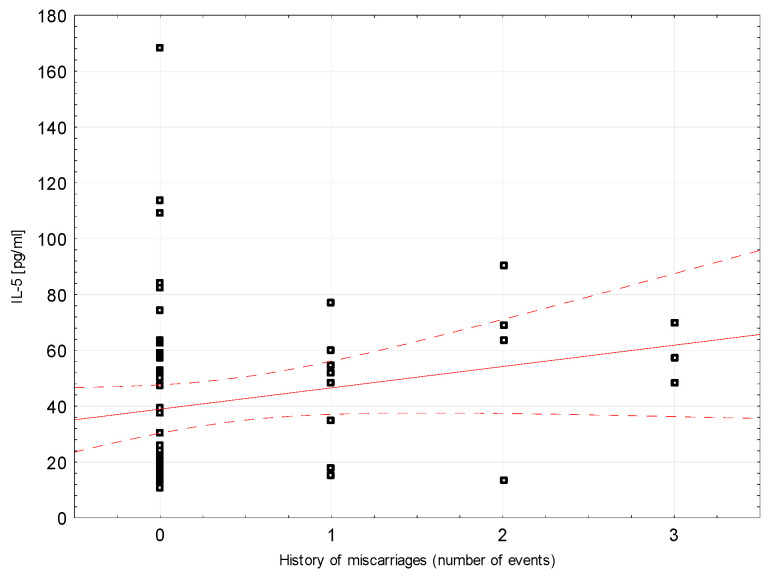
A scatter graph showing the correlation between the number of miscarriages and the concentration of IL-5 assessed in the blood serum of patients from the study group.

**Table 1 ijerph-18-08538-t001:** Comparison and distribution of demographic and clinical data in the study and control groups.

Variable	Study Group (*n* = 61) *N*; Me. ± SD. or Median (Range)	Control Group (*n* = 19) *N*; Me. ± SD. or Median (Range)	*p ^a^*	*p ^b^*
age (years)	30.35 ± 4.99	27.26; ± 3.05	0.0086	0.0466
hight (cm)	166.26 ± 5.34	166.10; ± 5.59	0.8919	0.1957
weight (kg)	67.00 (48.00–111.00)	58.00 (46.00–99.00)	0.0291	0.5777
BMI (kg/cm^2^)	23.59 (17.63–41.27)	21.23 (16.70–31.96)	0.0385	0.2971
Primipara yes no	37 (60.66%) 24 (39.34%)	11 (57.89%) 10 (52.63%)	0.6839	*N*/*a*
Multipara yes no	24 (39.34%) 37 (60.66%)	10 (52.63%) 11 (57.89%)	0.6839	*N*/*a*
1 miscarriage in anamnesis yes no	8 (13.11%) 53 (86.89%)	*N*/*a*	*N*/*a*	*N*/*a*
2 miscarriages in anamnesis yes no	4 (6.56%) 57 (93.44%)	*N*/*a*	*N*/*a*	*N*/*a*
>2 miscarriages in anamnesis yes no	3 (4.92%) 58 (95.08%)	*N*/*a*	*N*/*a*	*N*/*a*
gestational age (hbd)	9 + 4 (6 + 0 − 13 + 5)	10 + 3 (7 + 4 − 13 + 3)	0.2396	0.7130

^a^—univariate analysis, ^b^—analysis of covariance, *N*/*a*—not applicable.

**Table 2 ijerph-18-08538-t002:** Comparison of the levels of tested cytokines between the study and control groups.

Variable	Study Group (*n* = 61)					Control Group (*n* = 19)	*p ^a^*	*p ^b^*
	Primiparas (*n* = 37)	Multiparas (*n* = 24)	*p ^a^*	*p ^b^*	Total			
INF-γ [pg/mL]	2.36	12.30	0.8419	0.7180	2.36	2.02	0.6962	0.9779
Il-1β [pg/mL]	2.20	1.89	0.3634	0.4635	2.20	1.27	0.1320	0.3511
Il-6 [pg/mL]	1.49	1.42	0.5647	0.8942	1.44	1.53	0.8431	0.4256
TNF-α [pg/mL]	5.73	3.72	0.1438	0.5569	5.17	6.32	0.5046	0.0899
Il-4 [pg/mL]	0.69	0.39	0.5848	0.9467	0.64	0.24	0.0155	0.8901
Il-5 [pg/mL]	48.75	21.91	0.2232	0.6775	30.19	16.65	0.0003	0.0008
Il-9 [pg/mL]	1.07	0.08	0.2587	0.5341	0.12	0.07	0.1227	0.2654
Il-10 [pg/mL]	2.98	2.87	0.1395	0.2493	2.98	2.98	0.7216	0.7988
Il-13 [pg/mL]	0.86	0.55	0.4426	0.4176	0.78	0.61	0.3716	0.4212
TGF-β1 [pg/mL]	6154.31	6752.08	0.1500	0.5490	6370.61	46,795.32	0.0560	0.1963

^a^—univariate analysis, ^b^—analysis of covariance (age as covariate).

**Table 3 ijerph-18-08538-t003:** Spearman rank correlation between selected demographic variables and serum cytokines evaluated in the study group.

Pair of Variables	N	Age		Weight		BMI		Number of Miscarriages		Gestational Age (hbd)	
		Rho	*p*	Rho	*p*	Rho	*p*	Rho	*p*	Rho	*p*
INF-ɣ [pg/mL]	61	−0.023	0.8599	0.066	0.6116	0.062	0.6324	−0.296	0.0204	−0.177	0.1730
Il-1ß [pg/mL]	61	−0.260	0.0452	0.268	0.0371	0.314	0.0138	−0.050	0.7015	0.055	0.6742
Il-6 [pg/mL]	61	0.068	0.6076	0.012	0.9274	0.028	0.8321	0.119	0.3599	0.054	0.6782
TNF-α [pg/mL]	61	−0.257	0.0478	0.230	0.0746	0.308	0.0157	0.087	0.5064	−0.020	0.8801
Il-4 [pg/mL]	61	−0.161	0.2187	0.160	0.2174	0.244	0.0577	0.130	0.3191	−0.002	0.9902
Il-5 [pg/mL]	61	−0.150	0.2531	0.128	0.3267	0.195	0.1315	0.264	0.0394	0.022	0.8641
Il-9 [pg/mL]	61	−0.109	0.4088	0.204	0.1151	0.264	0.0396	0.197	0.1283	−0.071	0.5841
Il-10 [pg/mL]	61	−0.196	0.1333	−0.176	0.1739	−0.082	0.5279	0.085	0.5144	−0.019	0.8835
Il-13 [pg/mL]	61	−0.176	0.1781	0.159	0.2218	0.178	0.1690	−0.153	0.2403	−0.137	0.2913
TGF-ß1 [pg/mL]	61	0.130	0.3214	0.158	0.2249	0.215	0.0954	0.071	0.5863	0.025	0.8456

**Table 4 ijerph-18-08538-t004:** Correlation of Spearman ranks between pro and anti-inflammatory cytokines evaluated in blood serum in control group.

Pair of Variables		Proinflammatory Cytokines							
		INF-γ [pg/mL]		Il-1ß [pg/mL]		Il-6 [pg/mL]		TNF-α [pg/mL]	
Anti-Inflammatory Cytokines	N	Rho	*p*	Rho	*p*	Rho	*p*	Rho	*p*
Il-4 [pg/mL]	19	0.454	0.0510	0.731	0.0004	−0.186	0.4452	0.160	0.5130
Il-5 [pg/mL]	19	0.195	0.4246	0.446	0.0556	−0.182	0.4562	0.129	0.5978
Il-9 [pg/mL]	19	−0.175	0.4741	0.129	0.5988	−0.414	0.0780	−0.071	0.7732
Il-10 [pg/mL]	19	0.161	0.5101	0.364	0.1254	−0.610	0.0055	0.500	0.0294
Il-13 [pg/mL]	19	0.188	0.4398	0.089	0.7157	−0.026	0.9145	0.205	0.4000
TGF-ß1 [pg/mL]	19	0.087	0.7236	0.148	0.5448	−0.121	0.6205	0.314	0.1903

**Table 5 ijerph-18-08538-t005:** Spearman’s rank correlation between pro and anti-inflammatory cytokines evaluated in blood serum in the study group.

Pair of Variables		Proinflammatory Cytokines							
		INF-γ [pg/mL]		Il-1ß [pg/mL]		Il-6 [pg/mL]		TNF-α [pg/mL]	
Anti-Inflammatory Cytokines	N	Rho	*p*	Rho	*p*	Rho	*p*	Rho	*p*
Il-4 [pg/mL]	61	0.395	0.0016	0.616	<0.0001	−0.094	0.4730	0.826	<0.0001
Il-5 [pg/mL]	61	−0.039	0.7670	0.176	0.1742	−0.187	0.1496	0.448	0.0003
Il-9 [pg/mL]	61	0.034	0.7959	0.298	0.0198	−0.286	0.0253	0.517	<0.0001
Il-10 [pg/mL]	61	0.276	0.0313	0.167	0.1983	0.259	0.0436	0.499	<0.0001
Il-13 [pg/mL]	61	0.375	0.0029	0.469	0.0001	0.013	0.9195	0.366	0.0037
TGF-ß1 [pg/mL]	61	0.103	0.4280	0.053	0.6852	−0.060	0.6464	0.359	0.0045

## Data Availability

Data available on request from the corresponding author.

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
