# Peer review of "Pro- and Anti-Inflammatory Cytokines in the First Trimester—Comparison of Missed Miscarriage and Normal Pregnancy"

_ijerph, 2021, doi:10.3390/ijerph18168538_

Round 1

Reviewer 1 Report

Minor

Why non-parametric test? The data was not normal ? Normality test was performed?

Polish the english... Repetition of words ate the same phrase.. Ie. Line 110

Major

The characterization (topic 4.1) could be better explained if the data were shown as boxplot representing each dot as na individual or even with barplots and standard deviation... (comparison control VS study group)

Where are the raw data ? I was not able to find it in the manuscript and did not find any supplementary material. Therefore, I could not further check the data.

By the way, it should be placed in the main text, since it 

Author Response

Dear Reviewer.

Thank you for your comments and suggestions.

Due to the non-normal distribution of all studied cytokines (assessed with use of  D`Agostino-Pearson test), nonparametric tests were used. This information has been added to the statiscical analysis section.

The work has been checked and improved in terms of the English language.

Suitable figures have been added to the text. The data is shown according to your suggestions.

I am really sorry I have not attached the tables and figures to the main text before. This is my fault. There is a correct version now.

Reviewer 2 Report

The results by Kwiatek et al. did not confirm that women with missed miscarriage compared to normal pregnancy had an advantage of any type of immune response. Overall, this is an interesting study. However, some issues must be clarified.

Specific comments:

-Kindly homogenize the word “anti-inflammatory” in the whole document.

-Page 2, line 76. It should say “uncomplicated”.

-Results section should be divided into 3 subsections: 1) Characteristics and Comparison of Selected Demographic and Clinical Variables in the Study and Control Groups; 2) Comparison of Values of Selected Cytokines in the Study and Control Groups and the Subgroups of the Test Group, 3) Correlations between pro-and anti-inflammatory cytokines.

Major comments:

The authors indicated that in the study group, women were significantly older, had higher body weight and higher BMI. How these covariates could affect the results of the cytokine concentrations?. For example, results confirm up-regulation of certain pro-and anti-inflammatory cytokines in obesity, please see doi: 10.1371/journal.pone.0121971. In addition, aging is related to proinflammatory cytokines, please see DOI: 10.1097/00062752-200105000-00001. In this sense, a covariate adjustment (BMI, age, weight) by ANCOVA should be performed to avoid these effects.

Some studies have reported changes in the concentrations of the IL-10 family of cytokines and IFN-γ from nonlabor to labor(hyper-inflammatory) states. Some comments should be added in the discussion section. Please see doi: 10.1159/000480734.

Author Response

Dear Reviewer,

Thank you for your comments and suggestions.

The word “anti-inflammatory” has been homogenized in the whole document.

Page 2, line 76. It should say “uncomplicated - it has been corrected.

You are right, results section should be divided into 3 subsections. There are 3 subsections now.

The paragraph about the influence of maternal age and body mass has been added.

Some information about Il-10 and IFN-γ has been added to the discussion.

Reviewer 3 Report

The authors concluded that women with missed carriage compared to normal pregnanc had not an advantage of any type of immune response. However, authors performed the analysis only ELISA, another analysis such as quantitative RT-PCR may have a possibility to reveal the differences between them. In addition, I could not find the Tables in this manuscript (PDF) unfortunately.

Conclusion section must be further improved since the description in this section is duplicated in abstract section.

Author Response

Dear Reviewer,

Thank you for your comments and suggestions.

You are right that other tests like RT-PCR could improve the study and give more information. We will think about other tests in the next study.

I am really sorry I did not include the tables and figures in the main text. It was my fault. There is a correct version now.

The conclusion section has been improved.

Round 2

Reviewer 1 Report

I believe the authors satisfactorily answered the concerns raised. Therefore, I support the manuscript to be published in its current form.

Reviewer 2 Report

The authors have successfully answered all questions and comments and therefore I recommend accepting the article for publication in present form.

Reviewer 3 Report

The conclusion section was improved in comparison with previous version.

I`m looking forward to your progress by another analysis such as RT-PCR in the next study.